# Visfatin Promotes the Metastatic Potential of Chondrosarcoma Cells by Stimulating AP-1-Dependent MMP-2 Production in the MAPK Pathway

**DOI:** 10.3390/ijms22168642

**Published:** 2021-08-11

**Authors:** Shih-Ya Hung, Chih-Yang Lin, Cheng-Chieh Yu, Hsien-Te Chen, Ming-Yu Lien, Yu-Wen Huang, Yi-Chin Fong, Ju-Fang Liu, Shih-Wei Wang, Wei-Cheng Chen, Chih-Hsin Tang

**Affiliations:** 1Graduate Institute of Acupuncture Science, China Medical University, Taichung 404022, Taiwan; shihyahung@mail.cmu.edu.tw; 2Department of Medical Research, China Medical University Hospital, Taichung 404022, Taiwan; 3Department of Pharmacology, School of Medicine, China Medical University, Taichung 404022, Taiwan; u9957651@cmu.edu.tw; 4Graduate Institute of Biomedical Sciences, China Medical University, Taichung 404022, Taiwan; u106210218@cmu.edu.tw (C.-C.Y.); u105305004@cmu.edu.tw (Y.-W.H.); 5Department of Sports Medicine, College of Health Care, China Medical University, Taichung 404022, Taiwan; d2326@mail.cmuh.org.tw (H.-T.C.); d1762@mail.bh.cmu.edu.tw (Y.-C.F.); 6Department of Orthopedic Surgery, China Medical University Hospital, Taichung 404022, Taiwan; 7Division of Hematology and Oncology, Department of Internal Medicine, China Medical University Hospital, Taichung 404022, Taiwan; d12604@mail.cmuh.org.tw; 8Graduate Institute of Basic Medical Science, China Medical University, Taichung 404022, Taiwan; 9Department of Orthopedic Surgery, China Medical University Beigang Hospital, Yunlin 651012, Taiwan; 10School of Oral Hygiene, College of Oral Medicine, Taipei Medical University, Taipei 110301, Taiwan; Jufangliu@tmu.edu.tw; 11Department of Medicine, MacKay Medical College, New Taipei City 252005, Taiwan; shihwei@mmc.edu.tw; 12Graduate Institute of Natural Products, College of Pharmacy, Kaohsiung Medical University, Kaohsiung 807378, Taiwan; 13Institute of Biomedical Sciences, Mackay Medical College, Taipei 104217, Taiwan; 14Division of Sports Medicine & Surgery, Department of Orthopedic Surgery, MacKay Memorial Hospital, Taipei 104217, Taiwan; 15Department of Biotechnology, College of Health Science, Asia University, Taichung 413005, Taiwan; 16Chinese Medicine Research Center, China Medical University, Taichung 404022, Taiwan

**Keywords:** visfatin, chondrosarcoma, metastasis, matrix metalloproteinase-2, AP-1

## Abstract

Chondrosarcoma is a malignant bone tumor that is characterized by high metastatic potential and marked resistance to radiation and chemotherapy. The knowledge that adipokines facilitate the initiation, progression, metastasis, and treatment resistance of various tumors has driven several in vitro and in vivo investigations into the effects of adipokines resistin, leptin, and adiponectin upon the development and progression of chondrosarcomas. Another adipokine, visfatin, is known to regulate tumor progression and metastasis, although how this molecule may affect chondrosarcoma metastasis is unclear. Here, we found that visfatin facilitated cellular migration via matrix metalloproteinase-2 (MMP-2) production in human chondrosarcoma cells and overexpression of visfatin enhanced lung metastasis in a mouse model of chondrosarcoma. Visfatin-induced stimulation of MMP-2 synthesis and activation of the AP-1 transcription factor facilitated chondrosarcoma cell migration via the ERK, p38, and JNK signaling pathways. This evidence suggests that visfatin is worth targeting in the treatment of metastatic chondrosarcoma.

## 1. Introduction

Chondrosarcoma occurs typically in cartilage-enriched bone (e.g., femur, tibia, or pelvis) [1,2] and has a high propensity to metastasize to distant organs [1]. High-grade chondrosarcomas are particularly prone to metastasize to the lungs [3,4], so it is essential to have treatments that prevent this phenomenon.

Metastasis involves the synthesis of proteolytic enzymes such as matrix metalloproteinases (MMPs), which are capable of degrading the extracellular matrix (ECM) and basement tissue [5,6]. Not only have significantly higher levels of MMP-2 production been recorded in human chondrosarcoma specimens than in normal cartilage [7], but increasingly higher levels of MMP-2 expression stimulate the invasion and metastatic potential of chondrosarcoma cells [7,8]. Notably, chemokine (C-C motif) ligand 3 (CCL3)-facilitated increases in MMP-2 synthesis in chondrosarcoma cells encourage their migratory abilities, while inhibition of MMP-2 expression abolishes this effect of CCL3 [7]. Thus, inhibiting MMP-2 expression would likely be a useful means of preventing chondrosarcoma metastasis.

Visfatin is a proinflammatory adipocytokine that is abundantly expressed in visceral fats. It elicits insulomimetic activity and reportedly regulates different cellular functions such as growth, migration, differentiation, and apoptosis in mammalian cells [9,10]. Significantly higher visfatin levels have been identified in patients with different cancers compared with cancer-free individuals [11], which is not surprising in light of the evidence revealing that visfatin expression is critical for several tumor-related processes including proliferation, angiogenesis, metastasis and drug resistance [12,13,14]. Recent evidence has documented a potentially important new role for visfatin in the context of metabolic disease, showing that visfatin upregulates ECM proteins including osteopontin (Opn), collagen type VI, MMP-2, and MMP-9 in 3T3-L1 pre-adipocytes, and induces *Opn* gene expression via PI3K, JNK, MAPK/ERK, and NOTCH1 [15]. However, no details are available as to the involvement of visfatin in chondrosarcoma metastasis. In this study, we found that visfatin promotes chondrosarcoma metastasis in vitro and in vivo. Visfatin also facilitates the migration of human chondrosarcoma cells through AP-1-dependent MMP-2 production in MAPK signaling pathways.

## 2. Results

### 2.1. Visfatin Facilitates MMP-2-Dependent Migration in Chondrosarcoma Cells

Visfatin facilitates the progression and survival of several different types of cancers [12]. MMP-2 reportedly regulates the invasion and metastasis of chondrosarcoma cells [7,16]. We initially found that visfatin treatment (10–100 ng/mL) promoted cellular migration ability of JJ012 and SW1353 cells, according to Transwell and wound healing assay data (Figure 1A–D). We also found that stimulating the cells with visfatin enhanced mRNA expression and protein synthesis of MMP-2 (Figure 1E,F). In addition, we found that mRNA expression and protein synthesis of MMP-2, as well as wound healing and cell migration ability, were similar when we treated chondrosarcoma cells with visfatin 50 or 100 ng/mL, so we chose 50 ng/mL in all further experiments. Transfecting the cells with MMP-2 siRNA diminished visfatin-induced promotion of cellular migration (Figure 1G–I), implying that MMP-2 is critical to the effects of visfatin upon chondrosarcoma cells.

### 2.2. The MAPK Signaling Pathway Mediates the Effects of Visfatin upon MMP-2 Synthesis and Migration of Human Chondrosarcoma Cells

The MAPK (ERK, p38, and JNK) signaling pathway is critical in chondrosarcoma metastasis [17,18]. Treating cells with ERK, p38, or JNK inhibitor (ERK II, SB203580, or SP600125) significantly antagonized visfatin-induced stimulation of cell migration and MMP-2 production (Figure 2A–C, Figure 3A–C and Figure 4A–C). Similar effects were observed when the cell lines were transfected with ERK, p38 or c-JNK siRNAs (Figure 2D–F, Figure 3D–F and Figure 4D–F). Visfatin stimulation time-dependently promoted ERK, p38, and JNK phosphorylation (Figure 2G, Figure 3G and Figure 4G).

### 2.3. AP-1-Mediated MMP-2 Expression Regulates Visfatin-Induced Stimulation of Chondrosarcoma Cell Migration

AP-1 transcriptional activity is critical for MMP-2-mediated cancer migratory activities [18,19]. AP-1 inhibitor treatment with tanshinone IIA or curcumin effectively inhibited visfatin-induced cellular migratory activities and lowered MMP-2 expression (Figure 5A–C). The same effects were seen when the cells were transfected with siRNA against c-Jun (Figure 5D–F), whereas visfatin stimulation facilitated the phosphorylation of c-Jun (Figure 5G). ERK, p38, and JNK inhibitors all suppressed visfatin-enhanced promotion of c-Jun phosphorylation and AP-1 luciferase activity (Figure 5H,I). Therefore, visfatin increases chondrosarcoma cell migration through AP-1-dependent MMP-2 production in the MAPK pathway.

### 2.4. Overexpression of Visfatin Facilitates the Metastasis of Chondrosarcoma Cells in the Mouse Lung

We used the orthotopic in vivo model of chondrosarcoma lung metastasis to examine the stimulatory effects of visfatin in metastatic chondrosarcoma [20]. JJ012/visfatin cells expressed high mRNA and protein levels of both visfatin and MMP-2, and they exhibited a high migratory ability (Figure 6A–E). JJ012/vehicle and JJ012/visfatin cells were orthotopically implanted into the right leg tibia and tumor size was monitored by the IVIS system (Figure 6F,G). Overexpression of visfatin significantly increased tumor growth in the tibia (Figure 6F,G). At 12 weeks, metastasis to the lung was significantly more likely with JJ012/visfatin cells than with JJ012/vehicle cells (Figure 6H,I). IHC results revealed significant increases visfatin and MMP-2 expression in the JJ012/visfatin orthotopic model (Figure 6J), confirming that visfatin facilitates the metastasis of chondrosarcoma to the mouse lung.

## 3. Discussion

Chondrosarcoma is a malignant bone neoplasm that constitutes almost one-third (~26%) of all bone cancers [21]. Chemotherapy and radiotherapy have very limited effectiveness, so treatment with surgery is therefore the major management modality for chondrosarcoma. This malignancy is notorious for its aggressive clinical course and propensity to metastasize [22]. An effective adjuvant remedy is urgently needed to suppress chondrosarcoma metastasis [1,3].

Visfatin, also known as nicotinamide phosphoribosyltransferase (NAMPT) or pre-B cell colony-enhancing factor (PBEF), is an adipocytokine with several intriguing properties. Visfatin essentially affects the biosynthesis of nicotinamide adenine dinucleotide (NAD) and causes NAD^+^ (the oxidized form of NAD) accumulation, thereby affecting many NAD-dependent proteins such as sirtuins, PARPs, MARTs, and CD38/157, promoting the synthesis of adenosine triphosphate (ATP) [10]. Such effects ultimately lead to drug resistance in cancer cells with enhanced proliferation and metastasis. Some research has reported that high levels of visfatin expression enhance NAD^+^ production and increase the proliferation of colorectal cancer cells by activating the Wnt/β-catenin pathway [23]. In gastric cancer, visfatin facilitates cell migratory and invasive abilities, as well as the EMT process, by targeting the SNAIL transcription factor SNAI1 via NF-κB signaling [24]. Thus, visfatin is critical for tumor cell proliferation, migration, and survival [24,25], although the effects of visfatin in chondrosarcoma metastasis are uncertain. Our cellular and preclinical investigations found that visfatin reliably promotes chondrosarcoma metastasis. We also confirmed that visfatin facilitates chondrosarcoma cell migration through AP-1-dependent MMP-2 expression in the MAPK signaling pathway.

Elevated MMP-2 levels are apparent in numerous tumor metastatic processes [26] and MMP-2 reportedly mediates chondrosarcoma metastasis [7,18]. Researchers have therefore proposed the targeting of MMP-2 in the management of tumor metastasis [26,27]. Here, we found that visfatin increases levels of mRNA and protein synthesis in chondrosarcoma cells. Genetic knockdown of MMP-2 reduced the effects of visfatin upon chondrosarcoma cell migratory activity. We also observed that overexpression of visfatin facilitated increases in MMP-2 synthesis and chondrosarcoma metastasis in vitro and in vivo, leading us to believe that visfatin promotes chondrosarcoma metastasis via the upregulation of MMP-2.

Activation of the MAPK pathway is important in the adjustment of different cellular effects [28,29]. This signaling pathway also regulates the expression of MMP-mediated cancer motility [30,31]. Our results show that visfatin promotes the phosphorylation of ERK, p38, and JNK, while their respective pharmacological inhibitors suppress visfatin-induced promotion of MMP-2 expression and chondrosarcoma cell migration. This phenomenon is confirmed by similar effects observed with genetic siRNAs of ERK, p38, and JNK. This evidence reveals that activation of ERK, p38, and JNK signaling controls visfatin-enhanced promotion of MMP-2 synthesis and migration of chondrosarcoma cells.

Numerous transcription factor-binding sites have been reported in the 5′-regulatory region of MMP-2 [32]. AP-1 is an important transcriptional factor that regulates MMP-2 transcriptional ability and tumor metastasis [33]. We found that both AP-1 inhibitors (tanshinone IIA and curcumin) inhibited visfatin-induced facilitation of MMP-2 synthesis and chondrosarcoma cell migration. Confirmation of these effects by genetic inhibition using c-Jun siRNA indicate that AP-1 transcriptional activation is involved in the effects of visfatin upon MMP-2 expression and chondrosarcoma cell migration. We also observed that visfatin enhances c-Jun phosphorylation and AP-1 luciferase activity. The pharmacological inhibitors of ERK, p38, and JNK all antagonized visfatin-mediated activities, suggesting that visfatin promotes AP-1-dependent MMP-2 production and chondrosarcoma migration through ERK, p38, and JNK signaling.

## 4. Materials and Methods

### 4.1. Materials

Visfatin, MMP-2, ERK, p38, JNK, c-Jun, and β-actin antibodies were obtained from GeneTex (Hsinchu, Taiwan). The phosphorylated forms of ERK, p38, JNK, and c-Jun antibodies were bought from Cell Signaling (Danvers, MA, USA). Taqman^®^ One-Step PCR Master Mix and qPCR primers and probes were bought from Applied Biosystems (Foster City, CA, USA). Recombinant human visfatin was obtained from PeproTech (Rocky Hill, NJ, USA). All other chemicals not already mentioned were acquired from Sigma–Aldrich (St. Louis, MO, USA).

### 4.2. Cell Culture

The human chondrosarcoma cell line SW1353 was bought from ATCC (Manassas, VA, USA), while the other chondrosarcoma cell line JJ012 was kindly provided by Dr. Sean P. Scully (University of Miami School of Medicine, Miami, FL, USA). JJ012 cells stably expressing the visfatin complementary DNA (cDNA) clone (JJ012/visfatin cells) were established according to our previous method [20]. Cells were cultured 50%/50% in DMEM/α-MEM, 10% FBS, and antibiotics, then maintained in a humidified incubator at 37 °C in 5% CO_2_.

### 4.3. Cell Migration Assay

Chondrosarcoma cells were cultured on the upper chambers of Transwell plates (Costar, NY, USA), while visfatin and pharmaceutical inhibitors were treated to the lower chambers. After 18 h of treatment, migrated cells were fixed with 3.7% formaldehyde and stained with crystal violet, finally counted manually under the microscope [34,35].

### 4.4. Wound Healing Assay

The confluent chondrosarcoma monolayer was scratched by a fine pipette tip to make extended scratches. Cells were then treated with the conditions as indicated, migratory activity was evaluated by microscopy after 24 h and the rate of wound closure was analyzed [36].

### 4.5. Western Blot

SDS-PAGE was used to resolve the extracted proteins, which were transferred to PVDF membranes, as described in our previous publications [37,38,39]. Membranes were blocked for 1 h with PBST containing 4% non-fat milk, then treated with antibodies targeting visfatin, MMP-2, and β-actin for 1 h, before being incubated for 1 h with HRP-conjugated secondary antibodies. We visualized the blot membranes using a Fujifilm LAS-4000 imaging system (GE Healthcare, Little Chalfont, UK).

### 4.6. mRNA and miRNA Quantification

Total RNA was extracted from chondrosarcoma cell lines using TRIzol kit (MDBio, Taipei, Taiwan) and RNA contents were examined by a NanoVue Plus spectrophotometer (GE Healthcare Life Sciences; Pittsburgh, PA, USA). The M-MLV RT kit (Thermo Fisher Scientific; Waltham, MA, USA) and the Mir-X™ miRNA First-Strand Synthesis kit were used to perform reverse transcription of total RNA into cDNA. Real-time PCR analysis was performed using SYBR with MMP-2 or GAPDH primers. GAPDH mRNA expression was used as an internal control. The specific primer sequences for these genes are as follows: MMP-2: 5’-GGCCCTGTCACTCCTGAGAT-3’ (forward), 5’-GGCATCCAGGTTATCGGGG A-3’ (reverse), GAPDH: 5’-AATGGACAACTGGTCGTGGAC-3’ (forward), and 5’-CCCTCCAGGGGATCTGTTTG-3’ (reverse). Quantitative real-time PCR (qPCR) analysis was examined according to our previous reports [40,41].

### 4.7. Luciferase Assay

Chondrosarcoma cells were transfected with the AP-1 luciferase plasmid (Stratagene; St. Louis, MO, USA) using Lipofectamine 2000, then stimulated with pharmaceutical inhibitors and visfatin for 24 h. Luciferase activity was monitored using a luciferase assay kit [40,42,43].

### 4.8. Small Interfering RNA (siRNA) Transfection

Knockdown efficiency was confirmed by wound healing, Transwell cell migration, real-time PCR, and Western blot assay, as described in the following sections. Cells were transfected with control, p38, JNK, ERK, c-Jun, or MMP-2 siRNAs, as according to the manufacturers’ recommendations in standard procedures. The siRNAs (ON-TARGET*plus* SMARTpool) were purchased from GE Dharmacon (Lafayette, CO, USA). Cells were transfected with siRNAs using Lipofectamine 2000 reagent [44].

### 4.9. Establishment of Stably Transfected Cells

JJ012/visfatin or JJ012/vehicle (control shRNA) plasmids were obtained from MDBio, Inc. (Taipei, Taiwan) and transfected into cancer cells using Lipofectamine 2000 transfection reagent. Twenty-four hours after transfection, stable transfectants were selected in G418 (Geneticin) (Life Technologies, Grand Island, NY, USA) at a concentration of 200 μg/mL. Thereafter, the selection medium was replaced every 3 days. After 2 weeks of selection in G418, clones of resistant cells were isolated.

### 4.10. Tumor Xenograft Study

JJ012/vehicle or JJ012/visfatin cells (5  ×  10^6^) were orthotopically injected into 4-week-old male BALB/c nude mice (Taipei’s National Laboratory Animal Center), according to a previous protocol [20]. Tumor growth in the tibiae was examined each week by bioluminescence imaging by Xenogen IVIS imaging system 200 (PerkinElmer, MA, USA). After 12 weeks, the mice were sacrificed by CO_2_ inhalation. The lungs were then excised for further examination. All animal procedures were approved and performed in accordance with the guidelines of the Institutional Animal Care and Use Committee of China Medical University (CMUIACUC-2019-079).

### 4.11. Immunohistochemistry (IHC) Staining

Mouse lung tissues were rehydrated and treated with primary anti-visfatin or MMP-2 antibodies, then incubated with biotin-labeled secondary antibody. The slides were treated with the ABC kit (Vector Laboratories, CA, USA) and then photographed using the microscope.

### 4.12. Statistical Analysis

All values are presented as the mean ± standard deviation (SD) Differences between two experimental groups were assessed for significance using the Student’s *t*-test and considered to be significant if the *p* value was <0.05.

## 5. Conclusions

In conclusion, our study has identified that visfatin facilitates the metastatic potential of chondrosarcoma cells via AP-1-dependent MMP-2 production in the ERK, p38, and JNK pathway (Figure 7). We believe that targeting visfatin expression in metastatic chondrosarcoma offers a new way to address this aggressive malignancy.

## Figures and Tables

**Figure 1 ijms-22-08642-f001:**
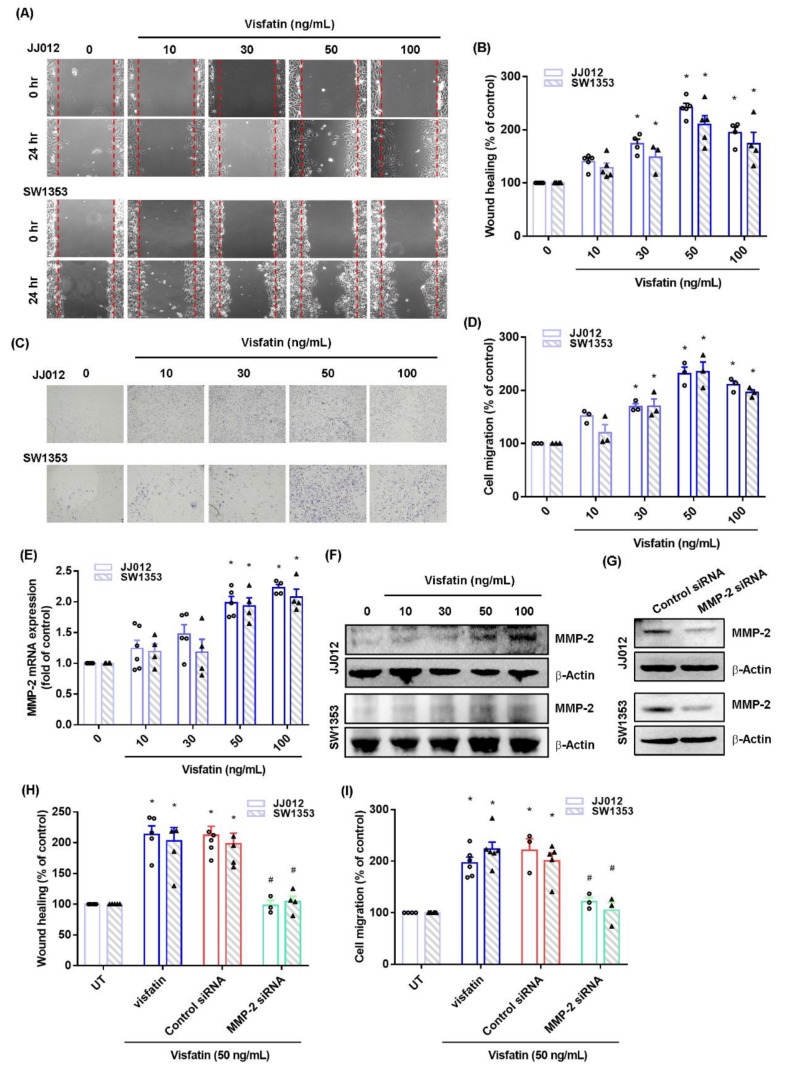
Visfatin promotes MMP-2-dependent cell migration in human chondrosarcoma. (**A**–**D**) Cells were incubated with visfatin (10–100 ng/mL) and cell migration was examined by Transwell and wound healing assays. (**E**,**F**) Cells were incubated with visfatin (10–100 ng/mL) and levels of MMP-2 mRNA and protein expression were examined by qPCR and Western blot assays. (**G**–**I**) Cells were transfected with MMP-2 siRNAs then stimulated with visfatin. Cell migration and MMP-2 expression levels were examined by Transwell, wound healing, and Western blot assays. Quantitative results are expressed as the mean ± SD. All experiments were repeated 3 to 5 times. * *p* < 0.05 compared with the UT group; # *p* < 0.05 compared with the visfatin-treated group. UT, untreated control.

**Figure 2 ijms-22-08642-f002:**
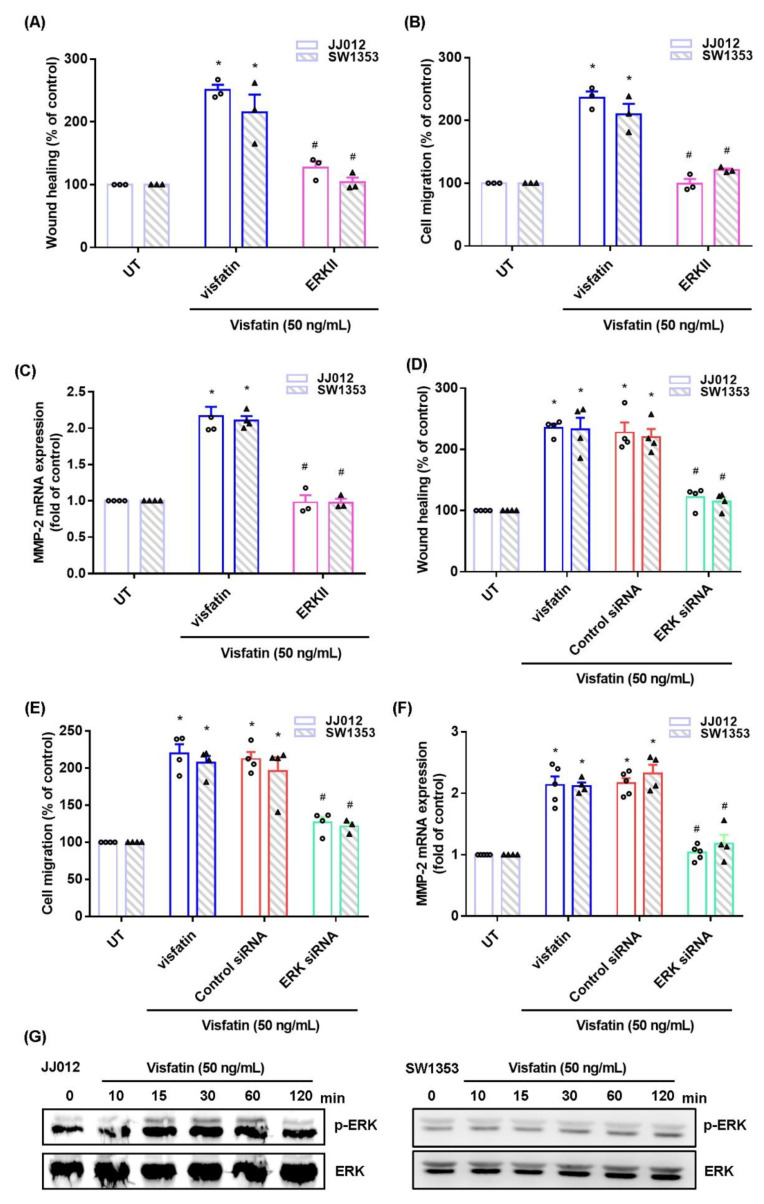
The ERK pathway mediates visfatin-induced MMP-2 expression and cell migration. (**A**–**F**) Cells were pretreated with an ERK inhibitor (ERK II) or transfected with an ERK siRNA, then stimulated with visfatin. Cell migration and levels of MMP-2 expression were examined by Transwell, wound healing, and qPCR. (**G**) Cells were incubated with visfatin for the indicated time intervals; ERK phosphorylation was examined by Western blot. Quantitative results are expressed as the mean ± SD. All experiments were repeated 3 to 5 times. * *p* < 0.05 compared with the UT group; # *p* < 0.05 compared with the visfatin-treated group. UT, untreated control.

**Figure 3 ijms-22-08642-f003:**
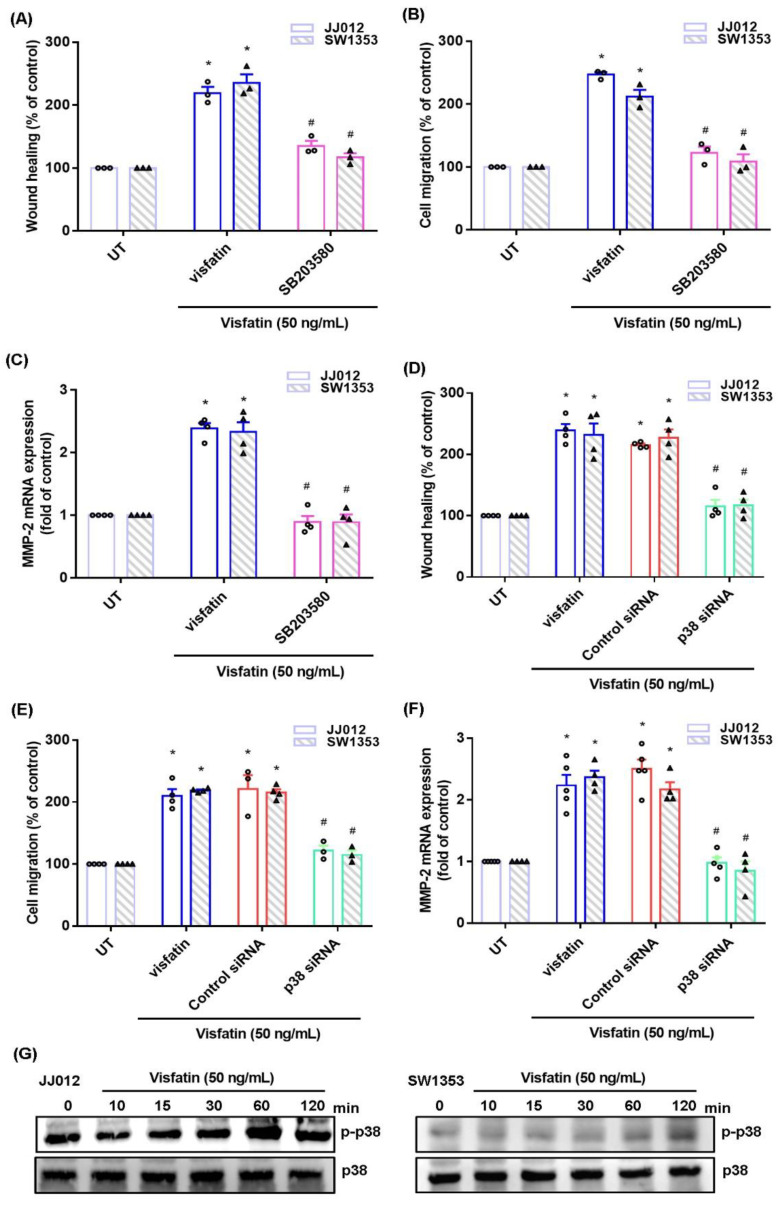
The p38 pathway mediates visfatin-induced MMP-2 expression and cell migration. (**A**–**F**) Cells were pretreated with a p38 inhibitor (SB253080) or transfected with a p38 siRNA, then stimulated with visfatin. Cell migration and levels of MMP-2 expression were examined by Transwell, wound healing, and qPCR. (**G**) Cells were incubated with visfatin for the indicated time intervals; p38 phosphorylation was examined by Western blot. Quantitative results are expressed as the mean ± SD. All experiments were repeated 3 to 5 times. * *p* < 0.05 compared with the UT group; # *p* < 0.05 compared with the visfatin-treated group. UT, untreated control.

**Figure 4 ijms-22-08642-f004:**
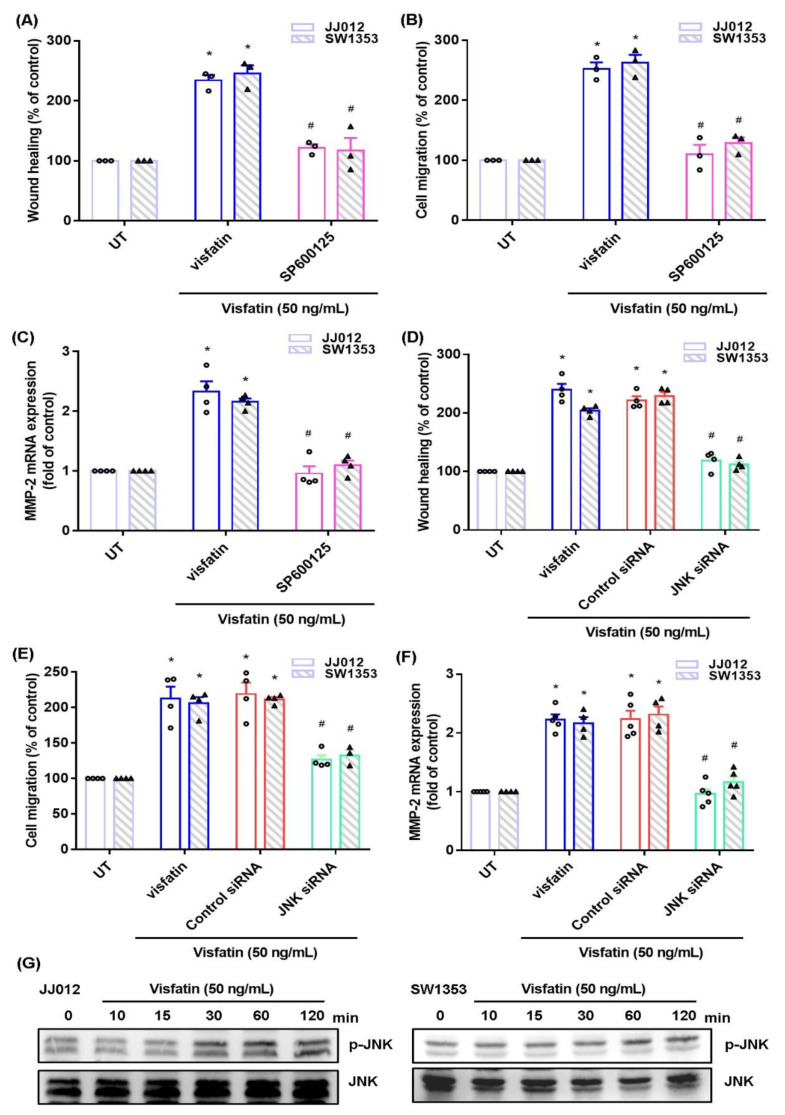
The JNK pathway mediates visfatin-induced MMP-2 expression and cell migration. (**A**–**F**) Cells were pretreated with a JNK inhibitor (SP600125) or transfected with a JNK siRNA, then stimulated with visfatin. Cell migration and levels of MMP-2 expression were examined by Transwell, wound healing, and qPCR. (**G**) Cells were incubated with visfatin for the indicated time intervals; JNK phosphorylation was examined by Western blot. Quantitative results are expressed as the mean ± SD. All experiments were repeated 3 to 5 times. * *p* < 0.05 compared with the UT group; # *p* < 0.05 compared with the visfatin-treated group. UT, untreated control.

**Figure 5 ijms-22-08642-f005:**
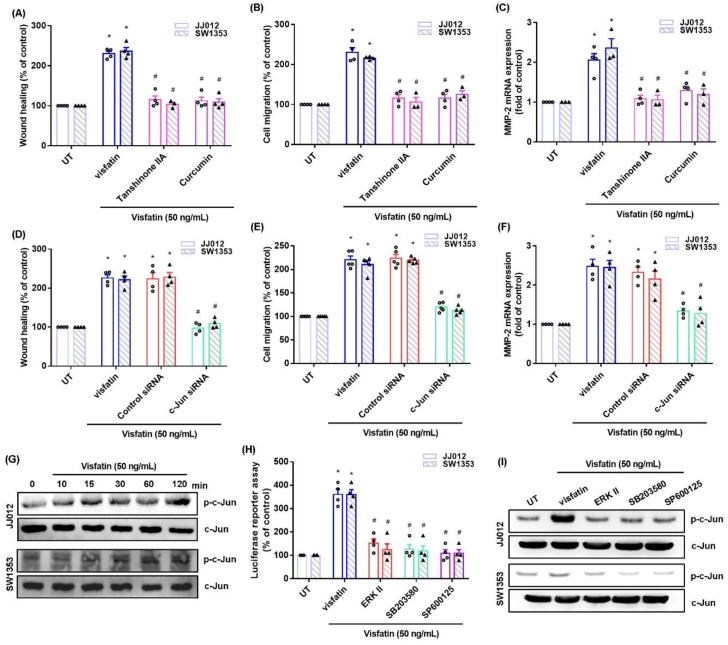
Visfatin induces MMP-2 production and cell migration through AP-1 activation in chondrosarcoma cells. (**A**–**F**) Cells were pretreated with AP-1 inhibitors (tanshinone IIA or curcumin) or transfected with a c-Jun siRNA, then stimulated with visfatin. Cell migration and levels of MMP-2 expression were examined by Transwell, wound healing, and qPCR. (**G**) Cells were incubated with visfatin for the indicated time intervals; c-Jun phosphorylation was examined by Western blot. (**H**,**I**) Cells were treated with indicated inhibitors then stimulated visfatin, the c-Jun phosphorylation and AP-1 luciferase activity was examined. Quantitative results are expressed as the mean ± SD. All experiments were repeated 3 to 5 times. * *p* < 0.05 compared with the UT group; # *p* < 0.05 compared with the visfatin-treated group. UT, untreated control.

**Figure 6 ijms-22-08642-f006:**
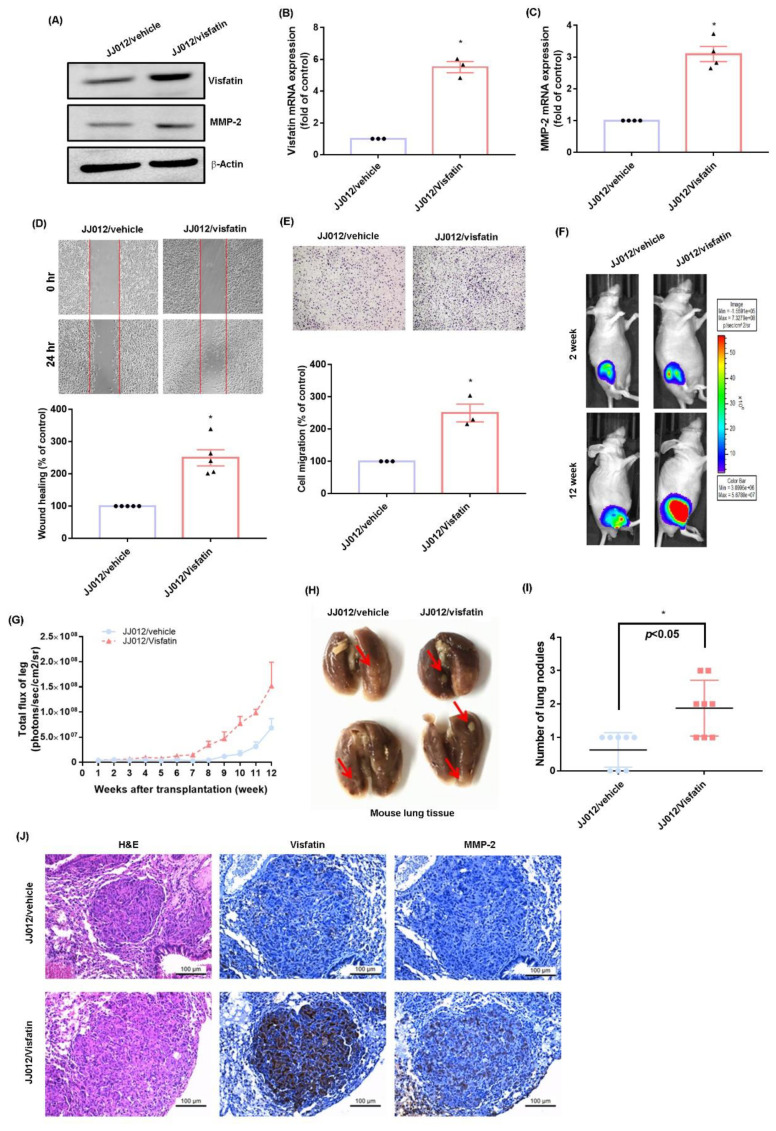
Visfatin promotes chondrosarcoma metastasis to lungs in vivo. (**A**–**E**) MMP-2 levels and migratory ability of JJ012/vehicle and JJ012/visfatin cells were examined by qPCR, Western blot, and Transwell. (**F**,**G**) The mice were injected with JJ012/vehicle or JJ012/visfatin cells. Lung metastasis was monitored by bioluminescence imaging at the indicated time intervals, then quantified by photon images. (**H**,**I**) After 12 weeks, the mice were humanely sacrificed and the lung tissue was excised, photographed, and quantified. (**J**) Levels of visfatin and MMP-2 expression in lung tumors were subjected to IHC analysis. Quantitative results are expressed as the mean ± SD. All experiments were repeated 3 to 5 times. * *p* < 0.05 compared with the JJ012/vehicle group. JJ012/vehicle, control shRNA; JJ012/visfatin, visfatin overexpression.

**Figure 7 ijms-22-08642-f007:**
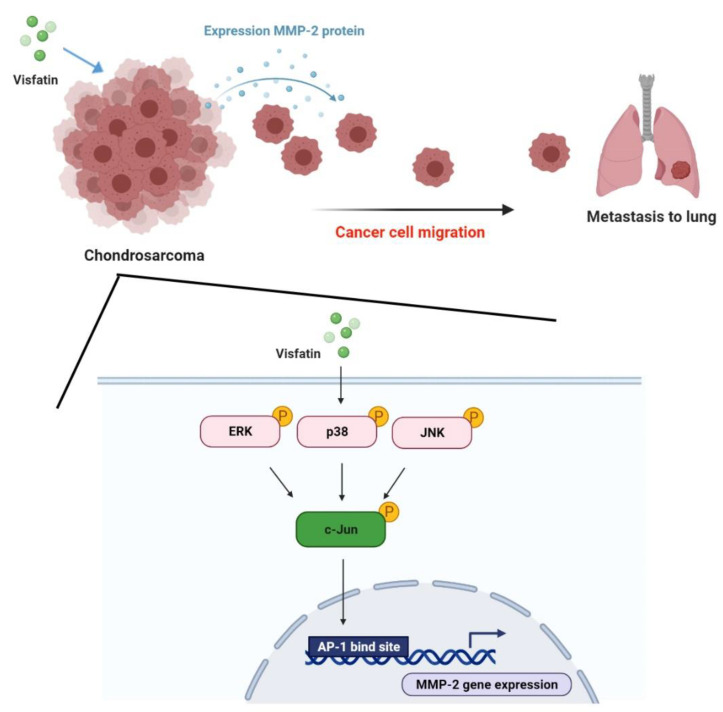
Schema illustrating the effects of visfatin in chondrosarcoma metastasis. Visfatin facilitates the metastatic potential of chondrosarcoma cells via AP-1-dependent MMP-2 production in the ERK, p38, and JNK pathways.

## Data Availability

The datasets used and/or analyzed during this study are available from the corresponding authors upon reasonable request.

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
