# Peer review of "Visfatin Promotes the Metastatic Potential of Chondrosarcoma Cells by Stimulating AP-1-Dependent MMP-2 Production in the MAPK Pathway"

_ijms, 2021, doi:10.3390/ijms22168642_

Round 1

Reviewer 1 Report

The topic of this research is "Visfatin promotes the metastatic potential of chondrosarcoma cells by stimulating AP-1-dependent MMP-2 production in the MAPK pathway". Two chondrosarcoma cell line SW1353 and JJ012 were used, and the research methods were Wound healing assay, Cell migration assay and Tumor xenograft study. The results of the study found that visfatin promoted the induction of MMP-2 synthesis through the activation of AP-1 transcription factors in the ERK, p38, and JNK signaling pathways. It was also found that overexpression of visfatin enhanced lung metastasis in a mouse model of chondrosarcoma.

Their experimental results support the research and conclusions. I believe that their experiments are sufficient and appropriate to draw the conclusions they are making. The research results have meaningful discoveries and academic value.

The explanations in the Introduction and discussions can be strengthened, and the relevant literature can be cited to strengthen the explanation of the effect of visfatin, especially for glucose uptake and cell proliferation, which affect cell migration and make the content of the paper more complete.

Author Response

Dear Reviewer,

We greatly appreciate the comments from your Reviewers on our manuscript, Visfatin promotes the metastatic potential of chondrosarcoma cells by stimulating AP-1-dependent MMP-2 production in the MAPK pathway (ijms-1309905). We have carefully revised the manuscript according to their suggestions, using red font to mark up the changes in our Word file. We have addressed their specific points in the text below.

Reviewer #1

Q: The explanations in the Introduction and discussions can be strengthened, and the relevant literature can be cited to strengthen the explanation of the effect of visfatin, especially for glucose uptake and cell proliferation, which affect cell migration and make the content of the paper more complete.

A: We thank the Reviewer for this suggestion and we have accordingly added details about the effects of visfatin in the Introduction and Discussion sections. (P2, lines 78-82; P9-10, lines 188-203)

“Recent evidence has documented a potentially important new role for visfatin in the context of metabolic disease, showing that visfatin upregulates ECM proteins including osteopontin (Opn), collagen type VI, MMP-2 and MMP-9 in 3T3-L1 pre-adipocytes, and induces Opn gene expression via PI3K, JNK, MAPK/ERK, and NOTCH1 [15].”

“Visfatin, also known as nicotinamide phosphoribosyltransferase (NAMPT) or pre-B cell colony-enhancing factor (PBEF), is an adipocytokine with several intriguing properties. Visfatin essentially affects the biosynthesis of nicotinamide adenine dinucleotide (NAD) and causes NAD+ (the oxidized form of NAD) accumulation, thereby affecting many NAD-dependent proteins such as sirtuins, PARPs, MARTs and CD38/157, promoting the synthesis of adenosine triphosphate (ATP) [10]. Such effects ultimately lead to drug resistance in cancer cells with enhanced proliferation and metastasis. Some research has reported that high levels of visfatin expression enhance NAD+ production and increase the proliferation of colorectal cancer cells by activating the Wnt/β-catenin pathway [23]. In gastric cancer, visfatin facilitates cell migratory and invasive abilities, as well as the EMT process, by targeting the SNAIL transcription factor SNAI1 via NF-κB signaling [24]. Thus, visfatin is critical for tumor cell proliferation, migration and survival [24, 25], although the effects of visfatin in chondrosarcoma metastasis are uncertain. Our cellular and preclinical investigations found that visfatin reliably promotes chondrosarcoma metastasis. We also confirmed that visfatin facilitates chondrosarcoma cell migration through AP-1-dependent MMP-2 expression in the MAPK signaling pathway.”

    Now that all feedback from your Reviewers has been attended to, we sincerely hope that this revised manuscript is suitable for publication in Cancers.

Best regards,

Chih-Hsin Tang, PhD.

Reviewer 2 Report

Shih-Ya Hung et al. present new results about the specific signaling pathways of nicotinamide phosphoribosyltransferase (visfatin), an enzyme regulating the nicotinamide metabolism, cell survival, and B-lymphocyte activation. They described that visfatin facilitates MMP-2-driven cell migration, promotes wound healing and MMP-2 mRNA expression in two chondrosarcoma cell types: JJ012 and SW1353. These effects were generated through at least three different signaling paths: the ERK, p38, and JNK converging in activation of transcription factor AP-1. Inhibitors of ERK, p38, and JNK all antagonized the before-mentioned effects. Visfatin also proved to facilitate the metastasis of implanted chondrosarcoma cells in the mouse lung. The authors used an orthotopic model of lung carcinoma metastasis and proved that visfatin-treated JJ012 cells implanted to the tibia produce significantly more metastatic nodules in the lung.

These results are new and valuable. The experimental design is clear and logical, and also, the content has biological importance. However, before publication, the manuscript needs some improvements:

  1. Materials and methods – the experiments are well illustrated, but the text does not explain every detail. The authors used for the positive control 50ng/mL visfatin treatment, but they do not describe what the vehicle was and whether they did perform any vehicle treatment for control cells or not. They should argue why did they choose the 50 ng/mL dose. There is a thorough description of mRNA quantification, but the RT-PCR method for MMP-2 expression is not given, nor the primers are specified. There is a lack of specifications regarding the transfection experiments, too. Providing these details is mandatory for the validation of the experiments.
  2. Experimental design and graphs – throughout Figures 1 to 6, the authors show data distribution in the case of cell migration, wound healing, and MMP-2 expression. However, they do not indicate the number of replicates for each experiment (analyzing the graphs, these could be 3-4) and do not specify what their columns show: mean ± SE, or mean ± SD? These details are mandatory for the interpretation of data.
  3. Some small orthographical mistakes should be revised, like the following sentences in the abstract:

- “visfatin facilitated matrix metalloproteinase-2 (MMP-2)-dependent cellular migration.”

- “chondrosarcoma cell migration was facilitated by the activation of the AP-1 transcription factor”.

If these details could be efficiently revised, I agree with the publication of the manuscript in the International Journal of Molecular Sciences.

Author Response

Dear Reviewer,

We greatly appreciate the comments from your Reviewers on our manuscript, Visfatin promotes the metastatic potential of chondrosarcoma cells by stimulating AP-1-dependent MMP-2 production in the MAPK pathway (ijms-1309905). We have carefully revised the manuscript according to their suggestions, using red font to mark up the changes in our Word file. We have addressed their specific points in the text below.

Reviewer #2

Q1: Materials and methods – the experiments are well illustrated, but the text does not explain every detail. The authors used for the positive control 50ng/mL visfatin treatment, but they do not describe what the vehicle was and whether they did perform any vehicle treatment for control cells or not. They should argue why did they choose the 50 ng/mL dose. There is a thorough description of mRNA quantification, but the RT-PCR method for MMP-2 expression is not given, nor the primers are specified. There is a lack of specifications regarding the transfection experiments, too. Providing these details is mandatory for the validation of the experiments.

A: We thank the Reviewer for this feedback, which considerably improves the Materials and methods section.

(i) We have now described the methodology for vehicle treatment in Materials and Methods section (2.9. Establishment of stably transfected cells):

“JJ012/visfatin or control shRNA (JJ012/vehicle) plasmids were obtained from MDBio, Inc. (Taipei, Taiwan) and transfected into cancer cells using Lipofectamine 2000 transfection reagent. Twenty-four hours after transfection, stable transfectants were selected in G418 (Geneticin) (Life Technologies, Grand Island, NY, USA) at a concentration of 200 μg/mL. Thereafter, the selection medium was replaced every 3 days. After 2 weeks of selection in G418, clones of resistant cells were isolated.” (P12, lines 292-298)

(ii) We have now described the reasons for our selection of the 50 ng/mL dose, as follows:

“In addition, we found that mRNA expression and protein synthesis of MMP-2, as well as wound healing and cell migration ability, were similar when we treated chondro-sarcoma cells with visfatin 50 or 100 ng/mL, so we chose 50 ng/mL in all further experiments.” (P2, lines 94-97)

(iii) We have provided details of the RT-PCR method using SYBR with MMP-2 or GAPDH primers, as follows:

“Real-time PCR analysis was performed using SYBR with MMP-2 or GAPDH primers. GAPDH mRNA expression was used as an internal control. The specific primer se-quences for these genes are as follows: MMP-2: 5'-GGCCCTGTCACTCCTGAGAT-3' (forward), 5'-GGCATCCAGGTTATCGGGG A-3' (reverse), GAPDH: 5'-AATGGACAACTGGTCGTGGAC-3' (forward), 5'-CCCTCCAGGGGATCTGTTTG-3' (reverse).” (P11, lines 273-278)

(iv) We have added methodology details about the transfection experiments in the Materials and methods section, as follows:

“Knockdown efficiency was confirmed by wound healing, Transwell cell migration, real-time PCR, and Western blot assay, as described in the following sections. Cells were transfected with control, p38, JNK, ERK, c-Jun, or MMP-2 siRNAs, as according to the manufacturers’ recommendations in standard procedures. The siRNAs (ON-TARGETplus SMARTpool) were purchased from GE Dharmacon (Lafayette, CO, USA). Cells were transfected with siRNAs using Lipofectamine 2000 reagent [44].” (P11-12, lines 285-291)

Q2: Experimental design and graphs – throughout Figures 1 to 6, the authors show data distribution in the case of cell migration, wound healing, and MMP-2 expression. However, they do not indicate the number of replicates for each experiment (analyzing the graphs, these could be 3-4) and do not specify what their columns show: mean ± SE, or mean ± SD? These details are mandatory for the interpretation of data.

A: We thank the Reviewer for this suggestion. We have accordingly added numbers of replicates and mean ± SD values in the Figure legends 1 through 6, as follows:

“Quantitative results are expressed as the mean ± SD. All experiments were repeated 3 to 5 times.”

Q3: Some small orthographical mistakes should be revised, like the following sentences in the abstract:

- “visfatin facilitated matrix metalloproteinase-2 (MMP-2)-dependent cellular migration.”

- “chondrosarcoma cell migration was facilitated by the activation of the AP-1 transcription factor”.

A: We thank the Reviewer for this careful attention to our text and we have accordingly revised the Abstract, as follows:

- “visfatin facilitated cellular migration via matrix metalloproteinase-2 (MMP-2) production”

- “activation of the AP-1 transcription factor facilitated chondrosarcoma cell migration via”

  Now that all feedback from your Reviewers has been attended to, we sincerely hope that this revised manuscript is suitable for publication in Cancers.

Best regards,

Chih-Hsin Tang, PhD.

Round 2

Reviewer 2 Report

The authors brought significant improvements to the manuscript, which now is eligible to be published.